# Evaluation of Vaccination Training in Pharmacy Curriculum: Preparing Students for Workforce Needs

**DOI:** 10.3390/pharmacy8030151

**Published:** 2020-08-20

**Authors:** Mary Bushell, Jane Frost, Louise Deeks, Sam Kosari, Zahid Hussain, Mark Naunton

**Affiliations:** Discipline of Pharmacy, Faculty of Health, University of Canberra, Bruce, Canberra 2617, Australia; Jane.Frost@canberra.edu.au (J.F.); Louise.Deeks@canberra.edu.au (L.D.); Sam.Kosari@canberra.edu.au (S.K.); Zahid.Hussain@canberra.edu.au (Z.H.); Mark.Naunton@canberra.edu.au (M.N.)

**Keywords:** immunization, vaccination, pharmacy student, pharmacy education, pharmacy curriculum, accreditation standards

## Abstract

Background: To introduce and evaluate a university vaccination training program, preparing final year Bachelor of Pharmacy (BPharm) and Master of Pharmacy (MPharm) students to administer vaccinations to children and adults in community pharmacy and offsite (mobile and outreach) settings. Methods: Final year BPharm and MPharm students were trained to administer intramuscular vaccinations to adults and children. The education program embedded in pharmacy degree curriculum was congruent with the requirements of the Australian National Immunisation Education Framework. The training used a mix of pedagogies including online learning; interactive lectures; and simulation, which included augmented reality and role play. All pharmacy students completing the program in 2019 were required to carry out pre- and post-knowledge assessments. Student skill of vaccination was assessed using an objective structured clinical assessment rubric. Students were invited to complete pre and post questionnaires on confidence. The post questionnaire incorporated student evaluation of learning experience questions. Results: In both cohorts, student vaccination knowledge increased significantly after the completion of the vaccination training program; pre-intervention and post-intervention mean knowledge score (SD) of BPharm and MPharm were (14.3 ± 2.7 vs. 22.7 ± 3.3; *p* < 0.001) and (15.7 ± 2.9 vs. 21.4 ± 3.2; *p* < 0.001) respectively. There was no difference between the BPharm and MPharm in the overall knowledge test scores, (*p* = 0.81; *p* = 0.95) pre and post scores respectively. Using the OSCA rubric, all students (*n* = 52) were identified as competent in the skill of injection and could administer an IM deltoid injection to a child and adult mannequin. Students agreed that the training increased their self-confidence to administer injections to both children and adults. Students found value in the use of mixed reality to enhance student understanding of the anatomy of injection sites. Conclusion: The developed vaccination training program improved both student knowledge and confidence. Pharmacy students who complete such training should be able to administer vaccinations to children and adults, improving workforce capability. Mixed reality in the education of pharmacy students can be used to improve student satisfaction and enhance learning.

## 1. Introduction

Vaccination and injection skills training has been taught in some Australian pharmacy degree curriculums since 2013 [1]. Indeed, training was being taught in pharmacy schools before pharmacists were administering vaccinations in the practice setting [1,2]. The rationale for this was that both the profession and pharmacy schools were anticipating regulation change to expand the scope of practice to enable pharmacist-administered vaccination [1]. Teaching and upskilling pharmacy students to vaccinate would enable a work-ready graduate. In 2014, Queensland became the first jurisdiction, outside a pilot program, to modify regulations to enable pharmacists to vaccinate [3]. Since then, regulations across all Australian states and territories have been modified to allow appropriately trained pharmacists to administer vaccinations to adults and more recently children aged 10 and over [4,5,6,7,8].

Many pharmacy students across Australia have now completed vaccination training embedded within their university degree; however, until March 2019, training was not formally recognized. That is, students would complete university vaccination training, and then, once registered (provisionally or fully, dependent on jurisdictional regulation), complete training delivered by an external provider (e.g., Pharmaceutical Society of Australia or Pharmacy Guild of Australia) to be certified competent to vaccinate [9,10]. This resulted in duplication of training for many early career pharmacists and an inherent lag time between original knowledge and skills development and administration of vaccines in the practice setting.

In March 2019, the Australian Pharmacy Council (APC), the body responsible for the accreditation of pharmacy education in Australia and New Zealand, published *the Standards for the Accreditation of Programs to Support Pharmacist Administration of Vaccines version 1.1* [11]. The APC amended the standard to enable pharmacy students enrolled in an accredited pharmacy degree program, to complete a vaccination training program delivered either within the degree program curriculum or via an external provider, during their period of study [11]. This change enabled universities to train and certify students to vaccinate. However, authorization to administer vaccinations is determined by state and territory legislation; at the time of writing, regulations preclude pharmacy student vaccinations in all Australian states and territories. However, the move by the APC to recognize vaccination training embedded in pharmacy degrees removes duplicity of vaccination training and enables students to be ready to vaccinate once they register.

The scope of practice of the Australian pharmacist vaccinator is constantly evolving to include more vaccinations and expand the age groups that pharmacists can vaccinate to. The eligible age of patients that pharmacists can vaccinate varies across jurisdictions. Interestingly, even within a state or territory, the eligible age to vaccinate differs between vaccines. From May 2020, appropriately trained pharmacists across all States and Territories can administer the influenza vaccine to children aged 10 and over [5,6,8,12]. In most jurisdictions, pharmacists can administer measles-mumps-rubella (MMR) and whooping cough (dTpa) to individuals 16 and over. While in Victoria, pharmacists can administer the MMR and dTpa vaccines to people aged 15 years and over [7]. There is a clear trend to lower the age limit eligibility and increase the type of pharmacist-administered vaccinations, improving accessibility and vaccine uptake. More recently, regulation has been modified to enable pharmacists to administer vaccines outside the pharmacy setting via both mobile and outreach services [7].

Therefore, it is appropriate that pharmacy students are trained and certified competent to deliver a vaccination service to both children (aged 10 and up) and adults. To date, most Australian pharmacy schools have integrated vaccination training into undergraduate and postgraduate pharmacy degrees, with a focus on administering vaccinations to adults [2,13]. The vaccination training program developed by the authors and evaluated in this paper, used the learning outcomes for the National Immunization Education Framework for Health Professionals [14]. This paper describes and evaluates the teaching and learning of vaccination training embedded in the pharmacy curriculum at the University of Canberra.

## 2. Materials and Methods

### 2.1. Development

#### 2.1.1. Alignment with The Framework and Accreditation Standards

A vaccination training program (VTP) was developed in line with the National Immunization Education Framework for Health Professionals (The Framework) [14]. This framework was designed to facilitate the development of nationally consistent, quality education programs for Australian Health Professionals, who are not medical practitioners, who want to be recognized as competent to administer vaccinations within their scope of practice. The university VTP adopted the core learning objectives and outcomes from The Framework, and then the teaching team adapted them to be pharmacy specific. To do this, the standards and guidelines specific to pharmacy (Professional Practice Standards, Practice Guidelines for the provision of immunization services within pharmacy) [15,16] were considered and integrated where appropriate.

Vaccination training has been embedded in the Bachelor and Master of Pharmacy degrees at the University of Canberra since 2016. The training, co delivered by pharmacists, pharmacy and nursing academics (all authorized immunizers), focused on teaching the knowledge and skills to administer vaccinations to adults. In 2019, to ensure that teaching and learning is congruent with contemporary pharmacy practice, this training was expanded to include content and skills assessment of injections to children. As the pharmacist vaccinators did not have, at that point, experience administering vaccinations to children, a nurse practitioner qualified to provide immunizations, delivered the content, theory, skills training, and assessment related to children.

#### 2.1.2. Teaching Methods

Pharmacists work as part of a broader health care team. The developed vaccination training program was taught via an interprofessional teaching team, which included pharmacist and nurse vaccinators. With reference to and consistent with the literature on pharmacy student vaccination training, there were a variety of educational pedagogies used to promote understanding and skill competency [13]. Teaching included both face-to-face (internal) and non-face-to-face learning opportunities and delivery of content. See Table 1. Students were given access to the online non-face-to-face content at semester commencement. This learning material could be completed by students in an asynchronous fashion prior to the intensive workshops. The face-to face content was delivered over four intensive whole day sessions. Students were taught the knowledge and skills to administer both IM and subcutaneous (SC) vaccinations and how to appropriately manage anaphylaxis.

#### 2.1.3. Simulation

To simulate environments and prepare students for real experience, the training program used the following: role-plays, mannequins, standardized patients, and mixed reality. Students had to role play and administer vaccinations to both a pediatric and adult low fidelity mannequin.

A mixed reality simulation technique using the Microsoft Hololens head-mounted devices along with the GIGXR applications Holohuman and Holopatient were used in the face -to-face delivery. The two applications were used to augment the students understanding of anatomy and physiology and to view a simulated patient who was portraying symptoms of anaphylaxis. Holohuman is an anatomy application that allows a student to gain a spatial understanding of anatomy and walk through the holographic body. As the student walks through the holographic image, layers of virtual anatomy peel away to reveal the underlying structures. This provided students with a unique way of identifying landmarks (i.e., deltoid muscle) for intramuscular (IM) vaccination. It was used to enable students to visualize the shoulder (synovial) joint and to recognize why a shoulder injury related to vaccine administration (SIRVA) would occur if given too high. Mixed reality has the power to engage the learner in a variety of interactive ways, which until this point have not been possible.

### 2.2. Assessment

#### 2.2.1. IM Injection Skill Assessment

Students skill competency was assessed using an objective structured skills assessment (OSCA). See Appendix A. An authorized immunizer assessed student skill competency to administer a vaccination to both an adult and child mannequin and provided feedback at the end of the assessment.

#### 2.2.2. Knowledge Assessment

Students completed identical pre- and post-knowledge assessments on the content taught on the topic of vaccination. Thirty questions assessed understanding of the topics taught. There were questions that assessed knowledge of the National Immunization Schedule, immunological principles of vaccination, vaccine preventable diseases, the different types of vaccines and how they elicit an immune response, current legislation and regulations related to pharmacist administered vaccination, vaccine cold chain, how to appropriately administer vaccines, documenting the vaccination service, and managing anaphylaxis. To enable matching of the pre- and post-vaccination knowledge tests, while enabling students to be deidentified, students had to provide an answer to questions, such as who was their first teacher and the day of the month they were born, on both the pre and post-tests.

#### 2.2.3. Student Evaluation and Feedback on Vaccination Training

All students completing the vaccination training, embedded in the unit Pharmacy Practice 4, were invited to participate in the evaluation of the training program by completing a hard copy questionnaire at the completion of the training. Participating in the evaluation questionnaire was voluntary and no payment or other incentive was provided. The questionnaire was developed by the authors of this paper. Questionnaires were face validated by pharmacy and nursing academics, all authorized vaccinators.

Each evaluation questionnaire included 19 questions that required students to rate their level of agreement on 5-point Likert scale (strongly agree to strongly disagree) and two free text questions. One question asked what the student liked about the vaccination training, the other how the vaccination training could be improved. Descriptive statistics were conducted. Free text responses were analyzed to identify repeating themes.

#### 2.2.4. Ethics Approval

All subjects gave their informed consent for inclusion before they participated in the study. The study was conducted in accordance with the Declaration of Helsinki, and the project was approved by the Human Research Ethics Committee of the University of Canberra (HREC17-138).

## 3. Results

### 3.1. Knowledge Assessment

In total, in 2019, 52 students completed the vaccination training. Of this, 34 (65.4%) were enrolled in the final year of BPharm and 18 (34.6%) were enrolled in the final year of the M Pharm degree. See Table 2. When combined, 19/52 (36.5%) had a current First Aid Certificate, 12/52 (23.1%) had a current mental health first aid certificate, and 48/52 (92.3%) were currently working in a pharmacy. See Table 3. There was no association between working in pharmacy, having a current first aid certificate and/or mental health first aid certificate and the mean knowledge score of the pre-test. The only statistically significant finding was that students who held a first aid certificate performed better than students who did not have a first aid certificate on the post-knowledge test (*p* = 0.014).

The mean pre-intervention knowledge score for the cohort was 14/30, while the post intervention knowledge score was 22/30. The difference in mean vaccination knowledge scores pre and post educational intervention was better (*p* < 0.001) with a large effect size (Cohens D = 0.75). See Table 2.

The results show that there was no statistically significant difference between the scores for the knowledge assessment between bachelor and master cohorts. BPharm students mean score pre-educational intervention was 15/30, and for Master of Pharmacy students it was 16/30. The mean score post-intervention was 23/30 for B.Pharm students, and 21/30 for M.Pharm students (*p* = 0.95).

### 3.2. Skills Assessment

Using the OSCA rubric, all students (MPharm and BPharm) completing the training program were identified as competent in the skill of injection. All students (*n* = 52) scored a yes against the 25 criteria of the OSCA rubric (Appendix A). All students could administer an IM deltoid injection to a child and adult mannequin.

### 3.3. Student Evaluation of the Training

All students (*n* = 52, 100%) either agreed (16/52, 31%) or strongly agreed (36/52, 69%) that the vaccination training enhanced their knowledge of vaccination. All students (*n* = 52, 100%) either agreed (6/52, 11%) or strongly agreed (46/52, 89%) that the practical session of administering a vaccine was useful/beneficial. All students (*n* = 52, 100%) either agreed (9/52, 17%) or strongly agreed (43/52, 83%) that the practical session increased their confidence to administer vaccinations. When asked *‘I feel confident that I know the correct vaccination technique for both adults and children’*, one student (1.9%) responded neutral, 29/52 (58%) agreed and 22/52 (42%) strongly agreed. Students voiced value in having the content delivered by an interprofessional teaching team, which included pharmacists and nurses.

A sample of students provided simple but positive comments like:

*“good teaching team”* Pharmacy student A.

### 3.4. Mixed Reality

Students were both satisfied and valued the integration of mixed reality in the vaccination training. Students voiced that it helped with the understanding of certain concepts, for example, shoulder injury related to vaccine administration (SIRVA). From the feedback evaluation form:

*“Walking into the holohuman was really neat. I liked that the layers of the human peeled away and it felt like I looking inside a human layer by layer. It helped my understanding not only of anatomy but the importance of making sure when administering an injection, I administer it in the right spot.”* Pharmacy student B.

## 4. Discussion

Students’ knowledge significantly increased post the educational intervention vaccination training. There was no difference between BPharm and MPharm student knowledge pre or post education intervention. This indicates that delivery of the training program in the final year of both degrees enables comparable understanding of the content and skills taught and a work ready graduate.

One finding was that students who had completed a first aid certificate, had higher post vaccination training knowledge scores. This finding is interesting as students did not have a higher mean pre knowledge test score. One possible reason for this is that students complete first aid training as adjunct training, that is, while it is recommended, it is not compulsory for students to complete. Students have gone above expectation to complete the training and have demonstrated commitment to continuing education. This attitude to study may be extrapolated to their commitment to the vaccination training and the larger unit in which the training is embedded. Many recent studies have shown that an individual’s grit, perseverance and passion for long term goals is associated with higher academic grades [17,18].

The training employed a range of teaching pedagogies to promote student understanding, skill competency and confidence. Simulation is a learner-centred educational pedagogy that facilitates learning by exposing the learner to a situation which is based on or mimics a real-life event. Simulation includes a broad range of activities. The use of simulation as an educational tool enables experiential learning and constructivism. It provides students with an opportunity to create their own meaning and co-construct knowledge in a safe environment, taking knowledge learnt from the lectures to application of the skill. Consistent with the literature, this teaching evaluation shows that student’s value and like to learn using simulation when being taught medical skills, such as vaccination [19].

Students learnt how to administer both an IM vaccination to a child and adult low fidelity mannequin. A recent study showed that the use of high-fidelity simulation mannequins as a teaching tool resulted in lower or equal student performance of clinical skills when compared to low-fidelity simulation mannequins [20]. The use of the different low fidelity mannequins enabled students to learn how to best position themselves to administer the vaccination safely at a 90-degree angle. Students were educated to be seated when administering a vaccine to a seated individual. Using appropriately sized mannequins showed students why vaccinations given by a standing immunizer to a seated individual are linked with increased risk of being administered too high. Using the different mannequins provided students with a safe environment to problem-solve and learn prior to practicing in the professional setting [21].

Mixed reality (MR) is an emerging technology in health care education [22]. Consistent with previous studies exploring student acceptability, students enjoyed and valued the integration of this teaching tool in the vaccination training [23]. To correctly identify the IM deltoid injection site, pharmacy students were taught to use anatomical markers. They were taught to create a ‘triangle’ over the individuals’ deltoid with their fingers, the centre of which the injection is administered. They were educated to locate the acromion process (shoulder tip) and place their index finger (or first finger) along it. Then to place the second and third finger down underneath the index finger (the third finger becomes the base of the triangle). The fourth finger is then opened to create a ‘side’ of the triangle. In the middle (not the top or the bottom) of the triangle, the injection should be made. Using MR enabled students to peel away the body to see these important anatomical markers, contextualizing and providing insight as to why the content was taught and the potential outcomes of incorrect vaccine administration. Students using the mixed reality anatomical software could visually see, using the 3D animation, that vaccines that are administered too low can be injected into the radial nerve, while vaccines that that are given too far to the side can cause damage to the axillary nerve. This highlights the importance of administering the vaccination into the correct area.

Role play enabled the students to step through the process of delivering the vaccination service and encouraged both peer learning and formative feedback. Using role play, students learnt to communicate appropriate, evidence-based information about benefits and the potential risks of vaccination and obtain valid consent. The active learning approach is widely used in higher education and allows learning across cognitive, psychomotor, and affective domains [24].

Pharmacists work as part of a broader health care team. One strength of the vaccination training program was that it used an interprofessional teaching team. In doing so, students were provided with an opportunity for interprofessional learning and practice. This strengthened program delivery and enabled students to see the value of interprofessional collaborative practice, which will continue in the practice setting [25].

While both internationally and nationally there are only a small number of studies published on the delivery and evaluation of pharmacy vaccination training in pharmacy schools, the results of this study are consistent with other published evaluations [2,26]. Across the studies, students’ value gaining new knowledge and skills and report confidence to administer vaccinations on completion of training [2,13,26]. Consistent with this study, the two Australian studies that outline the development of a pharmacy VTP also used national pharmacy standards to inform material [2,26]. In contrast, where this study reports adopting the learning outcomes from the National Immunization Education Framework for Health Professionals, the other studies did not. This makes direct comparison between the training programs difficult.

There are currently varied approaches to when vaccination training is embedded in degree programs. There are pharmacy schools that embed vaccination training in the earlier years of the curriculum, to enable students to see themselves as future ‘clinical’ health professionals [26,27]. In the published studies, students report satisfaction and skill acquisition. They also report enabling students to see that they will be touching patients as part of their future professional role when providing care [28]. Given pharmacy students in Australia, can now complete a university vaccination training program and, when jurisdictional regulation allows, vaccinate without needing to complete additional training by an accredited external provider, the best year level to embed vaccination training should be further researched. In doing so, an evidence-based and consistent decision can be made about the best year level to introduce the training. Further, to date, research in Australia has not directly compared the delivery, skill acquisition, confidence, and competence of students across different university vaccination training programs, this too should be further explored.

Students, like authorized immunizers, completed vaccination training that is congruent with the National Immunization Education Framework for Health Professionals and demonstrated competency. There is a case for jurisdictional regulations to be modified to enable pharmacy student-administered vaccinations. International research demonstrates that pharmacy students, under the supervision of a credentialled vaccinator, can administer vaccines safely [28,29]. The application of skill acquisition in the clinical setting improves students’ self-confidence [27,30]. Having pharmacy students ready to vaccinate on placement would enable a more work-ready graduate and a critical workforce that can be used to promote vaccination uptake. This skill is likely to be of greater use particularly in times when vaccination demand is high, for example a pandemic. If and when the Covid-19 vaccine is available, mass immunization is likely to be needed in a relatively short period of time to mitigate the spread of the disease and enable international borders to reopen. Pharmacy students could be used to improve workforce capability in Australia.

## 5. Conclusions

The vaccination training program described in this paper was embedded in the final year of MPharm and BPharm curriculum and enabled the same skill competency and knowledge acquisition across cohorts. The training program incorporated a suit of teaching methods, including mixed reality, which had high student acceptability. Aligning with the changed scope of practice for Australian pharmacists, pharmacy students learnt how to administer vaccinations to both adults and children. International research demonstrates that pharmacy students, under the supervision of a credentialled vaccinator, can administer vaccines safely. Given student competency and readiness after completing vaccination training, there is a case for jurisdictional regulations to be modified to enable pharmacy student-administered vaccinations in Australia.

## Figures and Tables

**Table 1 pharmacy-08-00151-t001:** Mode and type of content delivered to students.

Face-to-Face	Non-Face to-Face
Lectures Case studies Simulation Mixed reality Standardized patients Mannequins Role-play Skills assessment (OSCA)	Required readings Interactive e-training Videos Online quiz

**Table 2 pharmacy-08-00151-t002:** Vaccination knowledge pre and post-intervention (*n* = 52).

Course	Pre-Intervention Mean Knowledge Score (SD)	Post-Intervention Mean Knowledge Score (SD)	*p*-Value	Cohen’s D
BPharm (*n* = 34)	14.26 (2.70)	22.47 (3.26)	<0.001	-
MPharm (*n* = 18)	15.72 (2.89)	21.39 (3.16)	<0.001	-
Overall (*n* = 52)	14.07 (2.82)	22.09 (3.23)	<0.001	0.75

**Table 3 pharmacy-08-00151-t003:** Knowledge assessment scores.

Variable	Pre-Intervention Mean Knowledge Ccore (SD)	*p*-Value	Post-Intervention Mean Knowledge Score (SD)	*p*-Value
Course				
BPharm	14.3 (2.7)	0.81	22.7 (3.3)	0.954
MPharm	15.7 (2.9)		21.4 (3.2)	
First aid certificate				
Yes	14.7 (2.8)	0.85	22.00 (2.3)	0.014
No	14.8 (2.9)		22.15 (3.7)	
Mental health certificate				
Yes	14.00 (3.0)	0.809	21.6 (3.6)	0.451
No	15.00 (2.8)		22.3 (3.2)	
Currently working in pharmacy				
Yes	14.5 (2.8)	0.651	22.0 (3.1)	0.553
No	17.5 (2.4)		23.8 (4.3)	

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
