# Peer review of "Evaluation of Vaccination Training in Pharmacy Curriculum: Preparing Students for Workforce Needs"

_pharmacy, 2020, doi:10.3390/pharmacy8030151_

Round 1

Reviewer 1 Report

The paper “Evaluation of a vaccination training program developed for Australian pharmacy students: preparing students for evolving workforce needs” represents a very interesting topic in the current international situation. In general, the manuscript is well-written and has substantial merit. However, there are a few suggestions I think could improve the paper.

The Abstract presents relevant information. However, the authors must pay attention to the Results subsection. For example, the knowledge assessment consisted of 30 questions. Furthermore, because the authors use the phrase “In both cohorts student vaccination knowledge increased after the completion of the vaccination training program (pre-intervention mean knowledge score (SD) 14/25 (2.8); Post-intervention mean knowledge score 22 /25 (3.2) p<0.001)”, I am under the impression that they computed separate knowledge scores for both cohorts, pre- and post-intervention, which is not the case (there is only one knowledge score for the entire cohort, pre- and post-intervention). Furthermore, please be careful in reporting p-values. Additionally, in lines 192-193 of the manuscript, the authors mentioned that “Using the OSCA rubric all students (both M Pharm and B Pharm) completing the training program 192 were identified as competent in the skill of injection”. I find this result to be very important and the authors could include it in the abstract.

The Introduction is well structured and provides the necessary information. However, there are several aspects that are unclear:

Line 64: “duplicity of vaccination training”?

Lines 82-85: “This novel vaccination training program that has been developed, mapped and evaluated against the learning outcomes for the National Immunization Education Framework for Health Professionals, and includes training on administering vaccinations to children”. This phrase in unclear.

Lines 95-99: “The university VTP adopted the core learning objectives and outcomes from The Framework and then adapted them to be pharmacy specific. When developing the VTP the standards and guidelines specific to pharmacy (Professional Practice Standards, Practice Guidelines for the provision of immunization services within pharmacy) [13,14] were considered and integrated where appropriate”. How were the core learning objectives and outcomes pharmacy specific? Who decided what was appropriate?

The Materials and Methods section should provide more information regarding the VTP (given that one of the main objectives of this paper is to describe it). For example, how long did it take to complete the program and how was the curriculum structured? What type of vaccinations do students learn how to perform? The authors mentioned only the intramuscular (IM) vaccination. Additionally, how were the students chosen for the study? What were the eligible criteria? The authors must mention the statistical analyses they employed.

Lines 112-114: “With reference to and consistent with the literature on pharmacy student vaccination training, there were a variety of educational pedagogies used to promote understanding and skill competency.[11]”. The authors provided the reference, but it would be helpful to mention the specific educational pedagogies.

Table 1: the authors should be consistent in using either uppercase or lowercase.

Line 122: why low fidelity mannequin?

Lines 128-129: “As the student walks through the holographic image it peels away the to reveal the underlying structures”. Peels away the…?

The authors should be consistent in reporting the Results. For example, lines 174-176: “There was no association between working in pharmacy, having a current first aid certificate and/or mental health first aid certificate and the mean knowledge score of the pre-test”. The authors do not provide a p value, most likely because the association is not statistically significant. However, for lines 183-185, they provide the information: “The results show that there was no statistically significant difference between the scores for the knowledge assessment between bachelor and master cohorts. BPharm students mean score pre-educational intervention was 15/30, and for Master of Pharmacy students it was 16/30 (p = 0.81)”.

Line 189. The authors already have a table 1. Also, they should provide a description of the content. Currently, the description is “This is a table”.

Lines 192-194: how did the evaluator decide that all students were competent in the skill of injection? Appendix A presents 25 evaluation criteria/evidence with yes or no responses. How many “yes” and how many “no” were considered acceptable?

Line 201: Why “interestingly”?

The Discussion section, at this point, is mainly focused on teaching pedagogies (simulation, mannequin, mixed reality) rather than on the evaluation of the program (which is one of the main aims of the study, along with its description) and needs to be improved. For example, are there any other vaccination programs to compare against the current one? The authors mentioned that “Vaccination and injection skills training has been taught in some Australian pharmacy degree curriculums since 2013” (lines 37-38). Was their efficacy tested? Do those trainings have the same curriculum? Do universities have the liberty to choose the content of the training? The current VTP was assessed using an objective structured skills assessment and pre and post knowledge assessments on the content taught on the topic of vaccination.  Are these skills different from other vaccination programs targeting different students from different specialties (for example, nursing students)? Furthermore, there is no strengths and limitations section.

Lines 273-274: “International research demonstrates that pharmacy students, under the supervision of a credentialled vaccinator can administer vaccines safely”. Please provide references.

Conclusion

Lines 289-290: “International research demonstrates that pharmacy students, under the supervision of a credentialled vaccinator can administer vaccines safely”. The authors used the same phrase earlier (lines 273-274), but did not provide references.

Author Response

Reviewer 1

Many thanks for the detailed feedback and review. Please see attached.

Reviewer 2 Report

All abbreviations to be defined at the time of their first use.  MMR and dTpa are used and then in the next sentence define on page 2.

Line 86 – “embedded the pharmacy curriculum” change to “embedded in the pharmacy curriculum’

In Table 1, column 1 why isn't the first letter of all the first words capitalized?

As far as I am concerned there is no such terms as highly statistically significant and using this phrase in the same sentence with a p-value or confidence interval is redundant. 

Instead of highly statistically significant (p <0.001) I would suggest you used the “better (p < 0.001) with a large effect size....”

Please make sure the table numbers and references to tables are numbered correctly.  There at two tables labeled as Table 1. 

The second table one: page 5 line 189 has no title.  Instead, there is a placeholder where the actual title should be located. 

Is line 206 page 5 missing something?  A comment at this locate from a single student does not add anything to the manuscript.  the same problem occurs on line 213.  Why only one student?  If it is necessary, why not say a sample of the student positive comments is illustrated by.... 

Some references use capitals on the title of the article and others only capitalized the first word, proper nouns, and abbreviations.  The same format needs to be used throughout the reference list.

Reference #3 appears to be incomplete.

Author Response

Good morning reviewer 2, 

Many thanks for your detailed comments and suggestions. Please see attached.
